Discovery of two new weevil species of Pholicodes Schoenherr, 1826 (Coleoptera: Curculionidae: Entiminae) from eastern Turkey

Gültekin Levent 1 lgultekin@gmail.com
Gültekin Neslihan 2
1 Biodiversity Application & Research Center, Atatürk University , Erzurum , Turkey
2 Faculty of Agriculture, Department of Plant Protection, Iğdır University , Iğdır , Turkey
Hussein Mona
Electronic publication date: 2025 Apr 7
Publication date: 2025
Volume: 13
Electronic Location ID: e19026
Received 2024 Oct 9; Accepted 2025 Jan 29
Copyright: © 2025 Gültekin and Gültekin
Copyright year: 2025
Copyright holder: Gültekin and Gültekin
License: This is an open access article distributed under the terms of the Creative Commons Attribution License, which permits unrestricted use, distribution, reproduction and adaptation in any medium and for any purpose provided that it is properly attributed. For attribution, the original author(s), title, publication source (PeerJ) and either DOI or URL of the article must be cited.
License URL: https://creativecommons.org/licenses/by/4.0/

Keywords: Pholicodes, New species, Curculionidae, Anatolia, Turkey

Funding: Atatürk University Scientific Research Project Council (BAP) FAD-2024-13943 The study was supported by the Atatürk University Scientific Research Project Council (BAP) FAD-2024-13943 project. The funders had no role in study design, data collection and analysis, decision to publish, or preparation of the manuscript.

==============================
Weevils are beetles belonging to the superfamily Curculionoidea, known for their elongated snouts commonly named as rostrum. This superfamily is the most species-rich group in Insecta as well as Animalia kingdom, all they are considered phytophagous. The broad nosed weevil genus of Pholicodes Schoenherr, 1826 is solely Palearctic distribution with forty species. In this study, two new species Pholicodes artemisiae sp. nov. and Pholicodes hakkaricus sp. nov. are described from eastern Turkey. Morphological taxonomic characters are digitally illustrated. The new species Ph. artemisiae sp. nov. is associated with Artemisia plant and Ph. hakkaricus sp. nov. collected on Inula helenium L. in the habitat mountain slopes.

Introduction

The weevils, Curculionoidea (Insecta: Coleoptera) are a hyper diverse superfamily in animal kingdom with about 62,000 described and named species in the world (Oberprieler, Marvaldi & Anderson, 2007), and the species-rich group of highly specialized phytophagous insects (Korotyaev, 2000). According to “Catalogue of Palaearctic Coleoptera” Volume 7 & 8; Curculionoidea I & II (Löbl & Smetana, 2011, 2013), total number of species and subspecies distributed in Turkey is 1,726. Of these, 1,643 species distributed Asian part of Turkey and 83 species in European part. Total number of endemic species is 378 and endemism rate is 21.9%.

The herbivorous weevil subfamily is the most species-rich subfamily in the family, with about 14,000 known species and of these 3,500 species distributed in the Palearctic region (Korotyaev, Konstantinov & Volkovitsh, 2009). The Palearctic genus Pholicodes Schoenherr, 1826 (Coleoptera: Curculionidae: Entiminae: Brachyderini) is represented by 40 described species, with 10 species known from Turkey (Alonso-Zarazaga et al., 2023; Davidian, 1992; Pelletier, 2003). In the recent revision published by Davidian & Gültekin (2024), the species number is reached 14 with three new species descriptions and a new faunal record. In the same article, Davidian & Gültekin (2024) published a key to the species of Pholicodes for Turkey and neighboring territory of Transcaucasus with digital photographs of the habitus of adults and aedeagus.

During beetle diversity survey from most eastern Anatolia to western, many locations were investigated in 2023 and 2024. In two locations, two new species of Pholicodes were discovered, named and described in this present article.

Materials and Methods

Research territory and field exploration: Field investigations were conducted from eastern to western Turkey between 2023–2024. Specimens were collected using sweeping nets, Japan umbrella or by hand-picked individual collecting. Locations were numbered as TR23-01 to TR24-48, geographical data recorded via GPS for each location.

Morphological study: Dry adult specimens were placed overnight into the lukewarm water; genitalia were dissected and placed overnight into 10% KOH to macerate soft tissues, and finally cleaned with distilled water and 70% ethanol. Genitalia were placed in glycerin and examined under the compound microscope and dissection microscope.

Digital photographing: The photographs of the habitus of adults were taken using a Canon DSRL 6D camera connected with Leica Z16APO Macroscope, and processed using Canon EOS Utility software. For the photographing of genitalia structures, Zeiss Axio Imager A2 upright microscope with attached Canon DSRL 6D camera were used. Photographs were then assembled to plates by Adobe Photoshop CS 6.0.

Terminology. Morphological terminology follows that in Van den Berg (1972), Oberprieler, Anderson & Marvaldi (2014), and Lyal (2024).

Collection depositories: Specimens are deposited in the Biodiversity Science Museum, Atatürk University, Erzurum, Turkey (ABBM) and Zoological Institute, Russian Academy of Sciences, St. Petersburg, Russia (ZIN).

Abbreviations: A, antennomere; T, tarsomere; V, ventrite.

Electronic Publication and Life Science Identifiers: The electronic version of this article in Portable Document Format (PDF) will represent a published work according to the International Commission on Zoological Nomenclature (ICZN), and hence the new names contained in the electronic version are effectively published under that Code from the electronic edition alone. This published work and the nomenclatural acts it contains have been registered in ZooBank, the online registration system for the ICZN. The ZooBank LSIDs (Life Science Identifiers) can be resolved and the associated information viewed through any standard web browser by appending the LSID to the prefix http://zoobank.org/. The LSID for this publication is: urn:lsid:zoobank.org:pub:urn:lsid:zoobank.org:pub:1ECC66EB-01AA-4519-AA1F-FEA22FE7F5DC.

Results

Taxonomy

Genus Pholicodes Schoenherr, 1826

Type species Pholicodes plebejus Schoenherr, 1826

Diagnosis: The genus Pholicodes is characterized by the following characters: rostrum more or less conical, rostral dorsum convex or flat, antennal pterygia narrow or moderately widened, usually visible in dorsal view, eyes moderately convex, apical declivity of elytra as a rule sloping downwards, femora unarmed, tarsal claws fused, integument of the body densely covered by small size scales and short hairs, inner side of metatibia of the male at apical part with long and dense hairs, aedeagus strongly dorso-ventrally arched, laterally to the ostium usually with rows of the short setae, ventral side of the lamella of spiculum ventrale with median longitudinal carina or carina absent (Schoenherr, 1826; Davidian, 1992; Pelletier, 2003).

Subgenus Pseudopholicodes Davidian, 1992

Type species Brachyderes albidus Boheman, 1840

Subgenus includes three species (Ph. albidus, Ph. vittatus Schilsky, 1912 and Ph. problematicus Davidian, 1992) after Davidian (1992) and only one species after Catalogue of Palaearctic Coleoptera (Alonso-Zarazaga et al., 2023).

Diagnosis: The subgenus Pseudopholicodes is characterized by the following features: rostrum conical, rostral dorsum convex, antennal pterygia narrow, integument of the body densely covered by small size yellowish and brownish scales and short hairs, the inner side of metatibia of the male at apical 1/2 or 1/3 with long and dense hairs, aedeagus laterally to the ostium with rows of the short setae or without setae, ventral side of the lamella of spiculum ventrale without median longitudinal carina (Davidian, 1992).

Pholicodes artemisiae sp. nov.

Type material: Turkey: Erzurum Prov., Pasinler district, 3–4 km N of Çiçekli Village, N39°54′20″, E41°32′56″, 2,130 m, 18.7.2023, L. & N. Gültekin, 8♂, 7♀. Holotype (♂) and 12 paratype deposited in Biodiversity Science Museum of Atatürk University—ABBM (Erzurum, Turkey), two paratype in ZIN (St. Petersburg, Russia).

Etymology: The name “artemisiae” refers to food plant of the weevil.

Description, Male

Body elongated elliptical (Fig. 1), length 6.2–6.6 mm. Integument dark brown, antennae and tarsi pale chestnut brown. Surface of body covered densely with small suboval to subtrapeziodal brownish to whitish pale milky brown scales, which are micro-striated longitudinally. Dominating color brownish, paler scales condensed on interstriae 3, 5 reflecting interrupted stripes, more distinct and regular on 7–8. Antennae and inner surface of metafemora glabrous, scales smaller and sparser on tibiae, rather narrow whitish scales sparsely present on apical part of scape and tarsomeres 1–2. Seta-like decumbent scales scattered among suboval scales. Epistomal margin, latero-ventral margin of mandible, funicle, tibiae and tarsi with suberect hairs. These hairs stronger and longer on epistomal and mandibular margins, densely covered inner side of metatibia at the apical one third. The inner margin of the pro- and mesotibia with 1–2 sharp denticles among the seta-like hairs.

Figure 1 Male habitus of Pholicodes artemisiae sp. nov., holotype.

Head relatively short behind eyes, eyes elongated oval (Fig. 2C), moderately convex, expanding slightly outward of the head contour. Rostrum trapezoidal shape in frontal view (Fig. 2A), short and thick, 1.05–1.10× as wide as long at the base, apex 0.81–0.83× narrower than at base width. Interocular pit as a short longitudinal line. Epifrons convex at basal half, then going to be concave medially toward frons, frons depressed, epistomal margin upward V-shaped delineated with raised epistomal carina. Rostrum in lateral view, slightly curved, scrobe narrow, moderately deep in anterior half, superficial basally, slightly curved, reach to eyes, closed apically before apex. Scape long, 1.65× as long as rostrum, finely sinuate, gradually widening apically, distinctly widened toward apex. Funicle almost subequal length with scape, antennomeres filiform, sub-conically elongated, A1 thicker than A2, the latter 1.5× as long as A1, shortest antennomere A5 0.45x as long as A2. Antennal club fusiform, elongate, 3.0× as long as wide.

Figure 2 Rostrum of Pholicodes artemisiae sp. nov.

(A) Male, frontal view; (B) female, frontal view; (C) male, lateral view; (D) female, lateral view.

Pronotum sub-quadrate (Fig. 1), 1.04–1.14× as wide as long, expanding outwardly at apical one third, pronotal disc moderately convex, basal margin weakly rounded, anterior margin straight. Prosternum short, anterior margin slightly emarginate.

Elytra elongated elliptical (Fig. 1), 2.05–2.30× as long as wide, subparallel sided in basal half, slightly and roundly expanded at midpart, and gradually narrowed apically. Basal margin weakly emarginate; scutellum small, narrow trapezoidal. Elytra fused and wingless. 1st and 2nd striae not fused together at apex.

Legs moderately long, femora stout; tibiae subcylindrical; protibiae distinctly (Fig. 3A) and mesotibia moderately incurved apically with mucro. Metatibia (Fig. 3C) almost straight, its apical part on inner margin with long and dense hairs. Apical setal comb short, blackish, composed of spines, which are connected basally to each other. Tarsi moderately long, wide; T1 and T2 trapezoidal shape, T2 distinctly narrower and shorter than T1, T3 1.67–1.70× as wide as T2, T4 barely visible at the base of T5, T5 dorso-ventrally curved, gradually dilated apically, claws fine and short, fused basally, very slightly divergent at apex. Underside of tarsi covered with spongy pad.

Figure 3 Tibia of Pholicodes artemisiae sp. nov.

(A) Protibia of male; (B) protibia of female; (C) metatibia of male; (D) metatibia of female.

Abdominal ventrites elongated trapezoidal (Fig. 4A), weakly concave medially on V1–V2, transversely on V3–V4, narrowed distinctly along the margins of V3–V4 posteriorly. V5 swollen, somewhat wide trapezoidal looks like U-shaped.

Figure 4 Abdominal ventrite of Pholicodes artemisiae sp. nov.

(A) Male; (B) female.

Male terminalia and genitalia: Tergite 8 subtrapezoidal (Fig. 5A), almost U-shaped, sub-erect hairs distributed on and margins at the basal two third. In frontal view, penis elongated, sub-parallel sided at basal two third, suddenly and distinctly enlarged at apical one third, ends trapezoidal shape with obtuse apex (Figs. 5B, 5C). Lateral margins of the enlarged apical part with 5–6 erected short setae (Fig. 5B). In lateral view, penis strongly curved, gradually and slightly narrowed from midpart to apex (Fig. 5D). Apodeme of penis and tegmen thin, apodeme shorter than half length of penis. Tegmen widely ringed around penis basally. Spiculum gastrale thin (Fig. 5E), elongate with sclerotized apical plate.

Figure 5 Male terminalia and genitalia of Pholicodes artemisiae sp. nov.

(A) Tergite 8; (B) frontal view of penis at apical part; (C) frontal view of penis; (D) lateral view of aedeagus; (E) spiculum gastrale.

Female in dimorphism

Body elliptical (Fig. 6), slightly wider and short than male, the length 6.0–6.3 mm. Eyes and scrobe similar in male (Fig. 2D). Rostrum short and thick (Fig. 2B), 1.20–1.25× as wide as long at the base, at the base 1.04–1.09× wider as apex. Interocular pit slightly wider and deeper (Fig. 2B). Pronotum 1.14–1.20× as wide as long, elytra 1.70–1.75× as long as wide, moderately widened in the middle part. Protibia (Fig. 3B) and middle tibiae less incurved apically, mucro smaller on both tibiae, hind tibia slightly curved. The inner margin of pro- and middle tibiae with 3–4 sharp denticles among the seta-like hairs; inner side of hind tibiae without long and dense hairs (Fig. 3D). Abdominal ventrites (Fig. 4B) flattened medially on V1–V2, transversely on V3–V4, margins gradually narrowed posteriorly from middle part of V1 to the posterior corner of V5. V5 is narrow trapezoidal looks like V-shaped.

Figure 6 Female habitus of Pholicodes artemisiae sp. nov., paratype.

Female terminalia and genitalia: Tergite 8 narrowly trapezoidal (Fig. 7A), V-shaped, dorso-medial part less sectorized, posterior corner finely and sparsely setose. Gonocoxites elongated (Fig. 7B), rectangular, apex with a series short, relatively thick, erect seta-like hairs sorted around stylus. Stylus short (Fig. 7B), sub-cylindrical, apex with a group sub-erect long hair. Spiculum ventrale (Fig. 7C) consisting of a long apodeme and a triangular lamella, which is apically margined with setae. Spermatheca C-shaped (Fig. 7D), nodulus and collum swollen, cornu with obtuse apex.

Figure 7 Female terminalia and genitalia of Pholicodes artemisiae sp. nov.

(A) Tergite 8; (B) gonocoxites; (C) spiculum ventrale; (D) spermatheca.

Habitat and plant association: Specimens were collected under Artemisia plants on mountainous hills with rocky-stony soil with open vegetation dominated by Artemisia (Figs. 8A, 8B). It was observed that adults were feeding on the leaves, and ran rapidly to avoid capture.

Figure 8 Habitat and plant association of Pholicodes artemisiae sp. nov.

(A) Artemisia sp.; (B) collecting weevil by N. Gültekin.

Differential diagnosis: This new species belongs to the subgenus Pseudopholicodes, as shown by the trapezoidal structure of the rostrum and by the lamella of the spiculum ventrale lacking a distinct carina. A close relationship with Ph. vittatus can be inferred by the two species sharing the following features: apical part of antennal scape and 1–2 tarsomeres with narrow whitish scales; protibia of the male moderately emarginated at inner side; aedeagus with short setae in preapical part. Pholicodes artemisiae differs from Ph. vittatus and Ph. problematicus, by having a symmetrical apical half of the aedeagus, the aedeagus being asymmetrical in the latter two species. The new species is similar to Ph. problematicus in separate 1st and 2nd elytral striae at apex (anterior tibiae of the male of the latter are rather strongly emarginate at the inner side). Aedeagus of the Ph. albidus and Ph. problematicus in the preapical part without setae. According to the description of Ph. florae Pelletier, 2003, this species clearly differs from Ph. artemisiae sp. nov. by having a wider body and structure of aedeagus. The aedeagus of Ph. florae slightly and gradually dilated apically, ventral plate triangular at apex, anteapical hairs situated in the external corner (Pelletier, 2003).

Pholicodes hakkaricus sp. nov.

Type material: Holotype (♀), Turkey: Hakkâri Prov., 32.2 km NE of Hakkâri, Zap River Valley, N37°40′58.6″, E44°04′47.1″, 1,709 m, 25.5.2024, 1♀, M. S. Taylan leg. Holotype deposited in Biodiversity Science Museum of Atatürk University—ABBM (Erzurum, Turkey).

Etymology: The name “hakkaricus” refers to type locality of the province Hakkâri.

Description, Holotype, Female

Body oblong ovate (Fig. 9), length 8.0 mm. Integument black, antenna and tarsi blackish to dark chestnut brown. Integument of the body with subtrapezoidal scales, micro-striated longitudinally, pale milky brownish to creamy whitish. Among these scales, with stick form decumbent setae distributed densely. Inner surface of mesofemora and metafemora glabrous. Scales become smaller and sparser on femora and tibiae, rather narrow whitish scales sparsely present on apical part of scape, surface of scrobe and tarsomeres 1–2. Seta-like suberect hairs present on lateral side of epistomal margin, latero-ventral margin of mandible, funicle, tibiae and tarsi. These seta-like hairs somewhat longer on mandibulae, middle and metatibia. Inner margin of protibia with 7–8 sharp denticulate spins, meso- with 4–5 similar spins and metatibia with 5–6 respectively. Apical setal comb tibiae densely sorted, partly connate basally.

Figure 9 Female habitus of Pholicodes hakkaricus sp. nov., holotype, dorsal view.

Head very shortly visible behind eyes; eyes subconical (Fig. 10A), strongly convex, widest behind of middle, distinctly protrude out of the head contour, head across eyes almost as wide as anterior margin of pronotum. Basal half of rostrum in dorsal view (Fig. 10A) sub-trapezoidal, somewhat compressed laterally on the level of posterior margin of scrobe; sub-quadrate at apical half. Rostrum short and thick, 1.05× as wide as long, at apex 0.95× as wide as basal width. Epifrons canaliculated longitudinally (Fig. 10A), narrow and superficial at medial part, frons depressed, epistomal margin upward V-shaped, feebly carinated. Rostrum in lateral view, slightly curved, scrobe deep, gradually and strongly widened basally, at the base 3.50× as wide as apical part, curved with distinctly arched ventral margin downward, neither reached eyes nor close to eyes, closed anteriorly before apex. Scape long, 1.10× as long as rostrum, almost straight, gradually widening apically. Antenna long (Fig. 10B), funicle slightly shorter than scape, antennomeres filiform, subconically elongated, A1 thicker than A2, the latter one 1.10× as long as A1, shortest antennomere A5 0.38× as long as A2. Antennal club fusiform, elongate, 3.0× as long as wide.

Figure 10 Rostrum and antenna of Pholicodes hakkaricus sp. nov. (A) rostrum, frontal view; (B) antenna.

Pronotum sub-quadrate (Fig. 9), 1.35× as wide as long, distinctly expanding outwardly at middle part, pronotal disc moderately convex, basal margin almost straight, anterior margin straight. Prosternum short, anterior margin slightly emarginate.

Elytra oblong ovate (Fig. 9) in dorsal view, strongly convex in lateral view (Fig. 11B), 1.45× as long as wide, subparallel sided in basal half, slightly and gradually expanded around midpart, then gradually and roundly narrowed posteriorly. Basal margin weakly emarginate; scutellum small, narrow trapezoidal. Interstria 1, narrower than 2, the latter subequal width with 3–5, interstriae 6 longitudinally and deeply depressed making straight canals (Figs. 11A, 11B) starting behind humeri and reaching to the apical declivity. These two canaliculated interstria are distinctly visible in dorsal and lateral views (Figs. 11A, 11B).

Figure 11 Elytra of Pholicodes hakkaricus sp. nov.

(A) Lateral view; (B) dorsal view.

Legs moderately long, femora stout; tibiae subcylindrical; protibia (Fig. 12A) and mesotibia almost straight, very slightly incurved apically, metatibia curved inward and dorso-ventrally (Fig. 12B); protibia with small mucro, mesotibia with rather fine respectively. Tarsi moderately long, wide; T1 and T2 trapezoidal, T2 narrower and shorter than T1, T3 1.55× as wide as T2, T4 feebly distinguishable at the base of T5, T5 curved dorso-ventrally, gradually dilated posteriorly, claws small, fused basally, weakly divergent apically. Underside of tarsi with spongy pad.

Figure 12 Tibia of Pholicodes hakkaricus sp. nov.

(A) Protibia; (B) metatibia.

Abdominal ventrites trapezoidal, V2 depressed medially, V3–V4 weakly concave transversely; V5 trapezoidal V-shaped.

Female terminalia and genitalia: Tergite 8 subtrapezoidal (Fig. 13A), dorso-medial part less sclerotized, posterior margin setose. Gonocoxites elongated rectangular, in preapical part with abundant short and thick, erect setae (Fig. 13B). Stylus transverse, situated at apex. Spiculum ventrale (Fig. 13C) consist of long apodeme and subtriangle lamella rounded laterally, posterior margin densely setose. Spermatheca (Fig. 13D) C-shaped: cornu sickle-shaped, ramus very large, longer than collum. Structure of spermatheca of new species very similar with Strophomorphus iranensis Pelletier, 1999.

Figure 13 Female terminalia and genitalia of Pholicodes hakkaricus sp. nov.

(A) Tergite 8; (B) gonocoxites; (C) spiculum ventrale; (D) spermatheca.

Habitat and plant association: The type locality is on the western slope of the Zap River Valley, near a small creek running down the mountain slope. The dominant plant in the area where this weevil was found is Inula helenium L. (Figs. 14A, 14B).

Figure 14 Habitat and plant association of Pholicodes hakkaricus sp. nov.

(A, B) Habitat of Inula helenium L., dominating plant at type locality; (C) pre-blossoming Inula helenium L. in the habitat.

Differential diagnosis: This new species belongs to the subgenus Pholicodes s. str. based on rectangular shape of the rostrum. The new species is closely related to Ph. fausti (Reitter, 1890) with resemblance of body shape, rostrum and eyes. Easily differs from Ph. fausti by having straight lateral hollows along elytral interstriae 6, and by having the metatibiae curved dorso-ventrally. Apparently, presence of straight hollows along 6th elytral interstriae is unique character in the genus Pholicodes.

Taxonomic notes: Pholicodes fausti was described from Erzurum Province. Two known synonyms of Ph. fausti are Ph. oculatus Schilsky, 1912 and Ph. karacaensis (Hoffmann, 1954). The first one was described from “Syrien” (Schilsky, 1912) and the second one from “Montagnes de Karaca” [Şanlıurfa Prov.] (Hoffmann, 1954). Taxonomic position of this species is unclear, because in body structure and shape of the eyes similar to the genus Strophomorphus Seidlitz, 1867. This genus Strophomorphus is distinguished with the following features: the absence of a vertical depression at the posterior level of the vertex and the presence of the second antennomere of the funicle longer than the first. Based on the structure of aedeagus, the species “fausti” transferred to genus Pholicodes by Pelletier (1999).

Distribution: The geographical distribution of two new species are indicated on map (Fig. 15).

Figure 15 Distribution map two new species of Pholicodes in Turkey (★, Pholicodes artemisiae; ◼, Pholicodes hakkaricus).

Discussion

Eastern Turkey has a mountainous region, homeland to sources of big rivers such as Euphrates, Tigris and Aras, the highest mountain Ağrı (Ararat) and large lake Van. The region has high biodiversity harboring biogeographically Anatolian, Iran-Turan, Caucasian and Mesopotamian elements with high endemism rate plants and insect species. Despite this biological diversity, the biota of the region remains poorly known and faces several anthropogenic threats such as biotic degradation, desertification, salinization, overgrazing, erosion, habitat alteration, mis-foresting and agricultural applications, afforestation, alterations of river water regime based on dam constriction, urbanization, environmental pollution. Extreme examples of anthropogenic vegetation are overgrazed wormwood steppe and semidesert habitats. Rapid disappearance of the xerophilous complexes from the extraordinarily diversified and largely uninventoried Turkish biota makes preservation of the endangered plant and animal assemblages in different climatic zones of Turkey an urgent task (Korotyaev et al., 2016).

The weevil diversity investigations are ongoing our study topic from 25 years in this territory and a series of new species (approximately 50) recently described jointly (Korotyaev & Gültekin, 1999, 2001, 2003a, 2003b, 2020; Korotyaev, Gültekin & Colonnelli, 2002, 2017; Korotyaev, Gültekin & Gültekin, 2020; Dorofeyev, Korotyaev & Gültekin, 2004; Gültekin, 2005, 2006a, 2006b, 2008, 2013, 2022; Gültekin & Colonnelli, 2006; Davidian & Gültekin, 2006, 2007, 2015a, 2015b, 2016, 2022; Gültekin, Košťál & Gültekin, 2021; Gültekin & Korotyaev, 2011, 2012; Davidian, Gültekin & Korotyaev, 2017; Gültekin et al., 2008). These are evident reflecting richness of group and not yet well surveyed. Many of described new species are broad nosed weevil “Entiminae” group, similarly new finding Pholicodes species.

According to biogeographical data on Pholicodes (Alonso-Zarazaga et al., 2023; Davidian & Gültekin, 2024), Transcaucasia is one of the richest diversity centers of the genus with 26 described species among total 43 species. The second rich territory is Anatolia with 14 species, eight of them endemic. Seven species are distributed in Mesopotamia, two species Turkistan, one species Crete and one species Saudi Arabia (Alonso-Zarazaga et al., 2023; Davidian & Gültekin, 2024). With these two new species descriptions, Palearctic fauna will be represented 45 species and Turkish fauna 16. It means that one third of Palearctic fauna is distributed here, and second diversity center of the genus after Caucasus where about 25 species distributed.

Based on recent revision by Davidian & Gültekin (2024) following species are distributed in Turkey: Pholicodes albidus Boheman, 1840, Ph. aslani Davidian & Gültekin, 2024, Ph. brunneomaculatus Pelletier, 2003, Ph. elisabethae Pelletier, 2003, Ph. fausti (Reitter, 1890), Pholicodes florae Pelletier, 2003, Ph. lepidopterus Boheman, 1842, Ph. pancaucasicus Davidian, 1992, Ph. perdurus Reitter, 1895, Ph. pseudalbidus Davidian & Gültekin, 2024, Ph. pusillus Stierlin, 1885, Ph. stanislavi Davidian & Gültekin, 2024, Ph. theresae Pic, 1910, Ph. viridescens Reitter, 1900. Majority of the species are distributed in northeastern Turkey. Of these species, Ph. elisabethae is known only central Turkey, Ph. perdurus from middle Black Sea Region, Ph. fausti in SE Anatolia, Ph. viridescens in eastern Mediterranean. Upon look at vertical distribution of Turkish Pholicodes, it might be expressed that the group is mountain species and like usually high altitude between 1,100–3,200 m, majority of them prefer upper than 1,500 m, for example Ph. stanislavi at 2,105–3,200 m. Similarly, current two new species are collected on mountain at the elevation 1,700–2,130 m.

We would like to express our cordial thanks Dr. Genrik E. Davidian (All-Plant Protection Institute, St. Petersburg) for his valuable contributions on the early stage of manuscript. Professor Mehmet Sait Taylan (Hakkâri University) for support of the field investigation in Hakkâri province and his personally participating on the collecting trip. Many thanks Dr. Sci. Alexander S. Konstantinov (Smithsonian Institute, National Museum of Natural History, USDA–Systematic Entomology Laboratory, Washington), Dr. Sci. Mark G. Volkovich, Dr. Sci. Boris A. Korotyaev (Zoological Institute, Russian Academy of Sciences, St. Petersburg) and Professor Vladimir I. Dorofeyev (Komarov Botanical Institute (Herbarium), Russian Academy of Sciences, St. Petersburg) for collectively organizing field investigations in Turkey and their contributions development.

Additional Information and Declarations

Competing Interests

The authors declare that they have no competing interests.

Author Contributions

Levent Gültekin conceived and designed the experiments, performed the experiments, analyzed the data, authored or reviewed drafts of the article, and approved the final draft.

Neslihan Gültekin conceived and designed the experiments, performed the experiments, analyzed the data, prepared figures and/or tables, and approved the final draft.

Data Availability

The following information was supplied regarding data availability:

The data are available in the figures.

New Species Registration

The following information was supplied regarding the registration of a newly described species:

Publication LSID: urn:lsid:zoobank.org:pub:1ECC66EB-01AA-4519-AA1F-FEA22FE7F5DC.

Pholicodes artemisiae sp. nov. LSID: urn:lsid:zoobank.org:act:8D82678F-2790-4C6A-A2A8-9DC1FFAEE461.

Pholicodes hakkaricus sp. nov. LSID: urn:lsid:zoobank.org:act:EB5F9E16-FF18-4DB2-8232-3EA82E347BDA.

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
