# Peer review of "Discovery of two new weevil species of Pholicodes Schoenherr, 1826 (Coleoptera: Curculionidae: Entiminae) from eastern Turkey"

_PeerJ, doi:10.7717/peerj.19026_

## Round 0.1 · original submission · Major Revisions

· Academic Editor

Major Revisions

Based on reviewers’ opinion, the work lies within the journal scope and within the standards of the field. The descriptions are well-illustrated, and well-written. However, I believe that the paper would benefit from adding molecular data in addition to the morphometric characters (if available) and providing a location map for the new species. I wish that the authors consider depositing some specimens of Pholicodes artemisae discussing the *Pholicodes* of Turkey in a greater detail.

I recommend major revisions considering the comments of all reviewers.

·

Basic reporting

The peer-reviewed manuscript describes two new species of the genus Pholicodes from Eastern Turkey. The structure of this work is fully consistent with similar works on taxonomy. Literature well referenced and the authors reviewed all the main works on the genus Pholicodes. The presented photographs are of high quality and demonstrate not only the habitus, but also all their distinctive characters, as well as the genitalia and terminalia of males and females. I'm not a native English speaker and don't speak it professionally, so unfortunately I cannot fully verify this parameter. However, the text of this manuscript was completely understandable to me, so I'm sure that other readers will be able to understand it as well.
In my opinion, there are the following shortcomings:
1. It is necessary to bring the terminology to a single format. For example, in line 190 the term "protibia" is used to mean the fore tibia, but in the same line the middle tibia is called "middle tibia", instead of "mesotibia". This is repeated many times in all the descriptions and captions to the figures; I have marked some such examples in PDF. Also, the unequal terms “aedeagus” and “penis” are incorrectly used as synonyms (penis is only part of aedeagus, median lobe of the aedeagus) (line 498).
2. Since photos are an important part of modern taxonomic work, I would like to point out that some photos, especially Figures 7 and 13, are very dark and have a lot of extraneous garbage on them, which interferes with correct perception. For example, in Figure 7A there are dust particles that could be mistaken for sclerites. I would recommend that authors edit these images in a photo editor.

Experimental design

The methods are completely relevant. The authors follow the International Code of Zoological Nomenclature – types (holotype, paratype), their storage locations, differential diagnosis and other necessary information are clearly indicated. The descriptions of the new species are quite detailed and include all the necessary parts. The novelty of the taxa is convincingly demonstrated and leaves no doubt.

Validity of the findings

All underlying data have been provided.

Additional comments

I believe that after correcting the above-mentioned shortcomings, the article deserves to be published as soon as possible.

·

Basic reporting

this is interested study i made some changes to correct

Experimental design

good

Validity of the findings

good

Reviewer 3 ·

Basic reporting

This paper by Gultekin and Gultekin describes two new species of weevils from Turkey in the genus *Pholicodes*. The authors provide descriptions of each species, accompanied by habitus photographs and detailed photographs of other relevant anatomical structures. They also provide some guidance for how the species can be distinguished from other related taxa by providing differential diagnoses for each species.

The quality of English in the manuscript is reasonable, but many corrections will be necessary to elevate it to publication-quality standard. Although the meaning intended by the authors is always clear, there are several instances where the sentence formation doesn't conform to standard English grammar, which makes reading the text sometimes difficult. I have suggested changes in some instances in greater detail below.

Experimental design

This research is within scope of the journal and has been conducted within the standards of the field. The species descriptions are well-illustrated and the written descriptions are thorough. However, the two species descriptions are somewhat isolated and I believe that the paper would benefit from discussing the *Pholicodes* of Turkey in greater detail.

Overview of other species of *Pholicodes* in Turkey: While the information currently in the paper is sufficient for estabilishing the availability of these new names, I believe that there is an opportunity here for the authors to provide more tools to help readers appreciate the Turkish *Pholicodes* fauna better. To this end, I encourage the authors to consider adding an identification key to all 8 (10 if the number in line 39 doesn't included *P. artemisiae* or *P. hakkaricus*) species of *Pholicodes* known from Turkey. Ideally, this would be accompanied by habitus photographs of all species, and with any illustrations to help clarify any complicated descriptors in the key. Although this would require a fair amount of additional work on the part of the authors, I believe the result would be a more comprehensive and useful paper.

Distribution map: To assist readers who may be unfamiliar with the geography of Turkey, I recommend that the authors include a distribution map that shows the locations where the new species have been collected. If the authors choose to expand the paper a little to provide additional details about other Turkish *Pholicodes* species, their distributions could be added also.

Type specimens: I recommend that the authors consider depositing some specimens of *Pholicodes artemisae* in additional collections. This will have the benefit of making their

Validity of the findings

Unfortunately, I do not have access to any specimens of other species of *Pholicodes*, so I am unable to independently assess the characters the authors have used to distinguish these species. However, based on my understanding of other Entiminae genera, they seem to be appropriate. To reiterate my earlier opinion, I believe that providing a key to the *Pholicodes* of Turkey would be very useful for other workers wanting to identify specimens of the genus.

Additional comments

Line 36: Change "The herbivore weevil subfamily Entiminae is the richest group in the world with about ... " to "The herbivorous weevil subfamily is the most species-rich subfamily in the family, with about ... "

Lines 36 & 37: "14.000" and "3.500": Use of the full stop as a thousand seperator is not standard in English writing, and a comma would normally be used instead. The Journal will no doubt have a policy on this in their style guide which should be followed.

Line 39: Change "is represented 40 described species, eight species distributed in Turkey" to "is represented by 40 described species, with eight species known from Turkey".

Lines 42--52: Although this information about recent collecting efforts is interesting, it doesn't really impact on the results. Perhaps if you reported which of the 91 localities had *Pholicodes* and what species these were, perhaps on the map above, then it would be useful to know this information. But otherwise, much of this information is not paritcularly needed.

Line 59: "photographs of the habitus of adults and immature stages"---There are no immatures pictured in the paper, so the last three words of this extract can be deleted.

Line 105: "Subgenus *Pseudopholicodes* Davidian, 1992": Are both species described in this paper to be considered as being in this subgenus? The differential diagnosis of *P. hakkaricus*, comparing it with *P. fausti* (placed in the nominate subgenus, according to the "Cooperative Catalogue") suggests that this species should be in the subgenus *Pholicodes* and, if so, should be made explicit in this paper.

Line 197: Change "Tergite 8 narrow trapezoidal" to "Tergite 8 narrowly trapezoidal"

Line 199: Change "apex with a series short relatively thick" to "apex with a series of short, relatively thick"

Line 201: Change "Spiculum ventrale (Fig. 7C) consist of long apodeme and triangle lamella, which apical margin with setae" to "Spiculum ventrale (Fig. 7C) consisting of a long apodeme and a triangular lamella, which is apically margined with setae"

Lines 204--207: Change "Habitat and plant association: Specimens were collected under Artemisia plant on the mountain hill with rocky-stony soil and this open area mainly dominating Artemisia plant (Figs 8A-B). It was observed that adults were feeding with the leaves of this plant, running fast when trying to catch" to "Habitat and plant association: Specimens were collected under Artemisia plants on mountainous hills with rocky-stony soil with open vegetation dominated by Artemisia (Figs 8A-B). It was observed that adults were feeding on the leaves, and ran rapidly to avoid capture."

Lines 209--218: Change: "The new species belongs to the subgenus Pseudopholicodes in structure of rostrum and lamella of spiculum ventrale without distinct carina. It is closely related to Ph. vittatus in the following features" to "This new species belongs to the subgenus Pseudopholicodes, as shown by the structure of the rostrum and by the lamella of the spiculum ventrale lacking a distinct carina. A close relationship with Ph. vittatus can be inferred by the two species sharing the following features"
- Please state what aspect of the "structure of the rostrum" indicates an assignment within *Pseudopholicodes*
- Please state what feature of the "apical part of the antennal scape" is shared with *P. vittatus* (?by being abruptly widened)

Line 213: Change "From Ph. vittatus and Ph. problematicus differs in symmetrial apical half of aedeagus" to "Ph. artemisiae differs from Ph. vittatus and Ph. problematicus, by having a symmetical apical half of the aedeagus, the aedeagus being asymmetrical in the latter two species" (Please modify as appropriate).

Line 217: Change: "According description of the Ph. florae Pelletier, 2003 it is easily differing from Ph. artemisiae sp. nov. in wider body and structure of aedeagus." to "According to the description of Ph. florae Pelletier, 2003, this species clearly differs from Ph. artemisiae sp. nov. by having a wider body and [structure of aedeagus -- please describe the difference between these two species more fully. Ideally, the reader should not be assumed to be able to access Pelletier 2003 to be able to evaluate the aedeagal structure of P. florae]."

Lines 263--265: Change: "interstria 6 longitudinally and deeply depressed making straight canal (Figs 11A-B) starting behind humeri and to reach apical declivity. These two-straight canaliculated interstria distinctly visible in dorsal or lateral view (Figs 11A-B)." to "interstriae 6 longitudinally and deeply depressed making straight canals (Figs 11A-B) starting behind humeri and reaching to the apical declivity. These two canaliculated interstria are distinctly visible in dorsal and lateral views (Figs 11A-B)."

Lines 286--287: Change: "The dominant plant is Inula helenium L. in type locality where western slope of Zap River Valley with a small creak on mountain slope (Figs 14A-B)." to "The type locality is on the western slope of the Zap River Valley, near a small creek running down the mountain slope, The dominant plant in the area where this weevil was found is Inula helenium L. (Figs 14A-B)."

Lines 290--292: Change: "Easily differs from Ph. fausti in structure of elytra with two straight lateral hollows along of interstriae 6th and by hind tibiae which are curved dorso-ventrally." to "Easily differs from Ph. fausti by having straight lateral hollows along elytral interstriae 6, and by having the hind tibiae curved dorso-ventrally."

Line 297: Change: "Taxonomic position of this species is unclear ... similar to *Strophomorphus* .. closer to *Pholicodes*" to "The taxonomic position of this species is unclear ... similar to *Strophomorphus* .. closer to *Pholicodes*". Please also clarify that it is *P. fausti* that is somewhat similar to *Strophomophus*. Some comment about whether this also applies to *P. hakkaricus* would also be appropriate here.

Line 306: Change: "Although this richness, it is not well enough investigated yet and under anthropogenic pressure in several aspect." to "Despite this biological diversity, the biota of the region remains poorly known and faces several anthropogenic threats."
- Feel free to expand on what these threats are. The Discussion as a whole could benefit from greater consideration and expansion. What is the significance of these two weevil taxa? Are the plants they are found on range restricted? What factors may be limiting the distribution of the weevils?

---

## Round 0.2 · accepted · Accept

· Academic Editor

Accept

The authors have addressed all required corrections based on the reviewers' comments. the current version of the manuscript is ready for publication at PeerJ.

·

Basic reporting

The authors of the article have corrected all the main comments of the peer-reviewer, except for the comment regarding the presence of foreign particles particles in the photographs of some parts of the genital apparatus (for exemple, Fig. 7: A). I will leave this question to the discretion of the editors of the article. Otherwise, I believe that the article is ready and can be published in the journal.

Experimental design

no comment

Validity of the findings

no comment

Additional comments

no comment

·

Basic reporting

the revised version is improved and this paper in the current form can be accepted

Experimental design

good

Validity of the findings

good

Additional comments

no comments